# NAD+ enhances ribitol and ribose rescue of α-dystroglycan functional glycosylation in human FKRP-mutant myotubes

Carolina Ortiz-Cordero[1,2,3], Alessandro Magli[2,3], Neha R Dhoke[2], Taylor Kuebler[4], Sridhar Selvaraj[2], Nelio AJ Oliveira[2], Haowen Zhou[5], Yuk Y Sham[1,4], Anne G Bang[5], Rita CR Perlingeiro[1,2,3]*

[1]Department of Integrative Biology and Physiology, University of Minnesota, Minneapolis, United States; [2]Lillehei Heart Institute, Department of Medicine, University of Minnesota, Minneapolis, United States; [3]Stem Cell Institute, University of Minnesota, Minneapolis, United States; [4]Bioinformatics and Computational Biology Program, University of Minnesota, Minneapolis, United States; [5]Conrad Prebys Center for Chemical Genomics, Sanford Burnham Prebys Medical Discovery Institute, La Jolla, United States

**Abstract** Mutations in the fukutin-related protein (FKRP) cause Walker-Warburg syndrome (WWS), a severe form of congenital muscular dystrophy. Here, we established a WWS human induced pluripotent stem cell-derived myogenic model that recapitulates hallmarks of WWS pathology. We used this model to investigate the therapeutic effect of metabolites of the pentose phosphate pathway in human WWS. We show that functional recovery of WWS myotubes is promoted not only by ribitol but also by its precursor ribose. Moreover, we found that the combination of each of these metabolites with NAD+ results in a synergistic effect, as demonstrated by rescue of α-dystroglycan glycosylation and laminin binding capacity. Mechanistically, we found that FKRP residual enzymatic capacity, characteristic of many recessive FKRP mutations, is required for rescue as supported by functional and structural mutational analyses. These findings provide the rationale for testing ribose/ribitol in combination with NAD+ to treat WWS and other diseases associated with FKRP mutations.

*For correspondence:
perli032@umn.edu

Competing interests: The authors declare that no competing interests exist.

## Introduction

Mutations in the fukutin-related protein (*FKRP*) gene result in a broad spectrum of muscular dystrophy (MD) phenotypes, ranging from mild Limb-Girdle MD (LGMDR9) to Walker-Warburg syndrome (WWS), the most severe form of congenital MD (CMD) (*Beltran-Valero de Bernabé et al., 2004*; *Brockington et al., 2001a*; *Brockington et al., 2001b*). The biochemical hallmark of FKRP muscle disorders is hypoglycosylation of α-dystroglycan (α-DG), which leads to disruption in the interaction of α-DG with extracellular matrix proteins, in particular laminin-α2, which is essential for muscle fiber integrity (*Ervasti and Campbell, 1993*; *Ibraghimov-Beskrovnaya et al., 1992*). Due to its rarity and reduced life expectancy (<3 years), disease pathogenesis and treatment strategies remain elusive for WWS. To date, there is no effective treatment for FKRP-associated MDs (*Ortiz-Cordero et al., 2021*).

FKRP is a ribitol-5-phosphate transferase, that in tandem with fukutin (FKTN), adds ribitol 5-phosphate onto the 3GalNAc-β1-3GlcNAc-β1–4(P-6) Man-1-Thr/Ser modification of α-DG (Core M3) (*Manya et al., 2004*; *Yoshida-Moriguchi et al., 2013*), using cytidine diphosphate (CDP)-ribitol, which is produced by isoprenoid synthase domain-containing protein (ISPD) (*Gerin et al., 2016*; *Kanagawa et al., 2016*; *Praissman et al., 2016*; *Riemersma et al., 2015*). The presence of ribitol-5-

**eLife digest** Healthy muscles are complex machines that require a myriad of finely tuned molecules to work properly. For instance, a protein called alpha-DG sits at the surface of healthy muscle cells, where it strengthens the tissue by latching onto other proteins in the environment. To perform its role correctly, it first needs to be coated with sugar molecules, a complex process which requires over 20 proteins, including the enzyme FKRP. Faulty forms of FKRP reduce the number of sugars added to alpha-DG, causing the muscle tissue to weaken and waste away, potentially leading to severe forms of diseases known as muscular dystrophies.

Drugs that can restore alpha-DG sugar molecules could help to treat these conditions. Previous studies on mice and fish have highlighted two potential candidates, known as ribitol and NAD+, which can help to compensate for reduced FKRP activity and allow sugars to be added to alpha-DG again. Yet no model is available to test these molecules on actual human muscle cells.

Here, Ortiz-Cordero et al. developed such a model in the laboratory by growing muscle cells from naïve, undifferentiated cells generated from skin given by a muscular dystrophy patient with a faulty form of FKRP. The resulting muscle fibers are in essence identical to the ones present in the individual. As such, they can help to understand the effect various drugs have on muscular dystrophies.

The cells were then put in contact with either NAD+, ribitol, or a precursor of ribitol known as ribose. Ortiz-Cordero et al. found that ribitol and ribose restored the ability of FKRP to add sugars to alpha-DG, reducing muscle damage. Combining NAD+ with ribitol or ribose had an even a bigger impact, further increasing the number of sugars on alpha-DG.

The human muscle cell model developed by Ortiz-Cordero et al. could help to identify new compounds that can treat muscular conditions. In particular, the findings point towards NAD+, ribose and ribitol as candidates for treating FKRP-related muscular dystrophies. Further safety studies are now needed to evaluate whether these compounds could be used in patients.

phosphate is essential for the subsequent addition of $-3Xyl\alpha1-3GlcA\beta1$ and $(3Xyl\alpha1-3GlcA\beta1)$n- and, therefore, for allowing $\alpha$-DG to bind to extracellular membrane ligands (*Inamori et al., 2012*; *Manya et al., 2016*; *Praissman et al., 2014*; *Willer et al., 2014*). The addition of the pentose phosphate pathway (PPP) metabolites ribitol or ribose, a precursor of ribitol (*Gerin et al., 2016*; *Huck et al., 2004*), to fibroblasts from patients with ISPD mutations resulted in increased CDP-ribitol levels and rescue of $\alpha$-DG functional glycosylation. However, rescue varied significantly among samples with different mutations (*Gerin et al., 2016*; *van Tol et al., 2019*). *Cataldi et al., 2018* documented that ribitol treatment in FKRP mutant mice modeling severe LGMDR9 partially restored $\alpha$-DG functional glycosylation. More recently, *Nickolls et al., 2020* showed that ribitol treatment of embryoid bodies from an LGMDR9 patient-specific induced pluripotent stem (iPS) cell line partly recovered the glycosylation defect. Taken together, these results raise several possibilities: (i) ribitol and ribose may also have a beneficial effect in WWS associated with FKRP mutations, (ii) rescue by PPP metabolites might be mutation-specific, and (iii) other metabolites may potentiate the recovery of $\alpha$-DG functional glycosylation.

Another metabolite of interest is $\beta$-nicotinamide adenine (NAD+). NAD forms (NAD+, NADH, NADP, and NADPH) are essential cofactors for oxidoreductases in the PPP (*Singh et al., 2017*). Moreover, *Bailey et al., 2019* reported that NAD+ supplementation in FKRP-deficient zebrafish led to decreased muscle degeneration and improved muscle function when administered at gastrulation, before muscle development occurs. Nevertheless, to date, no human model exists to test or validate the therapeutic potential of these or any other compounds for WWS.

In this study, we took advantage of the ability of iPS cells to differentiate into skeletal myotubes (*Selvaraj et al., 2019b*) to establish a novel WWS patient-specific in vitro model. Our results demonstrate that this system recapitulates the major skeletal muscle hallmarks of WWS. Moreover, we find that ribitol and ribose can partially rescue functional glycosylation of $\alpha$-DG, and that administration of NAD+ along with each of these PPP metabolites significantly potentiates $\alpha$-DG functional glycosylation rescue.

## Results

### Impaired α-DG functional glycosylation in WWS iPS cell-derived myotubes

Using an integration-free approach, we generated iPS cells from a 1-year-old WWS male patient (FP4) harboring two mutations in exon 4 of the FKRP gene, dc.558dupC (p.A187fs) and c.1418T>G (p.F473C) (*Kava et al., 2013*). FP4 iPS cells express pluripotency markers, display normal karyotype, and develop teratomas containing cell types from all three germ layers (*Figure 1—figure supplement 1*). Using inducible expression of PAX7 (*Darabi et al., 2012*), we differentiated FP4 and control wild type (WT) iPS cells into myogenic progenitors and subsequently into terminally differentiated myosin heavy chain (MHC)-positive myotubes (*Figure 1A*). Immunostaining for MHC showed similar differentiation between WT and mutant FP4 myotubes (*Figure 1A*, upper panel). Staining with IIH6, a monoclonal antibody specific to the laminin binding domain of α-DG (*Ervasti and Campbell, 1993*), showed drastically reduced IIH6 immunoreactivity in FP4 myotubes (*Figure 1A*, lower panel), which was corroborated by western blot. In accordance with the loss of α-DG functional glycosylation, WWS FP4 myotubes showed decreased molecular weight for α-DG core (*Figure 1B*), marked reduction in IIH6 (*Figure 1B and C*), and most importantly, lack of laminin binding, as demonstrated by the laminin overlay assay (LOA) following enrichment by wheat germ agglutinin (WGA) pull-down (*Figure 1B*).

As proof of concept, we introduced a WT FKRP transgene into FP4 cells to determine whether WT FKRP could restore functional glycosylation of α-DG. FKRP-overexpressing FP4 myogenic progenitors gave rise to MHC-positive myotubes (*Figure 1—figure supplement 2A*) that displayed increased FKRP expression (*Figure 1—figure supplement 2B*) and enhanced immunoreactivity to

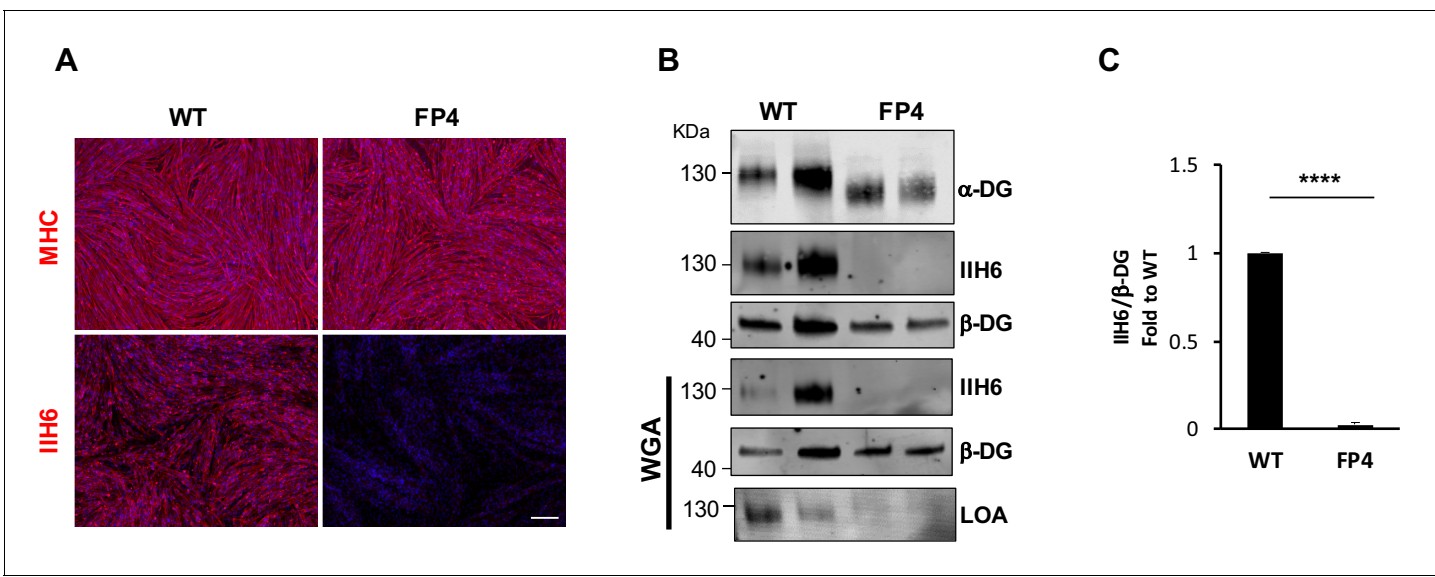

**Figure 1.** Patient-specific Walker-Warburg syndrome (WWS) induced pluripotent stem (iPS) cell-derived myotubes display reduced functional glycosylation of α-dystroglycan (α-DG). (A) Representative immunostaining of wild type (WT) and FP4 iPS cell-derived myotubes for myosin heavy chain (MHC) and IIH6 (in red). DAPI stains nuclei (in blue). Scale bar, 200 μm. (B) Representative western blot for α-DG core and α-DG functional glycosylation (IIH6) in WT and FP4 myotubes. β-DG was used as loading control. Lower panel shows wheat germ agglutinin (WGA) pull-down for these samples and representative laminin overlay assay (LOA) of WGA elutes shows laminin detection only in WT samples. (C) Graph bars show respective quantification of IIH6 (B) normalized to β-DG and shown as the fold difference of WT. Error bars represent standard errors of five independent experiments. Significance was evaluated by the unpaired Student's t test. ****p<0.0001.

The online version of this article includes the following figure supplement(s) for figure 1:

**Figure supplement 1.** Pluripotency characterization of reprogrammed FP4 induced pluripotent stem (iPS) cell line.

**Figure supplement 2.** Fukutin-related protein (FKRP) overexpression rescues functional glycosylation of α-dystroglycan (α-DG) in Walker-Warburg syndrome (WWS) FP4 induced pluripotent stem (iPS) cell-derived myotubes.

IIH6 (*Figure 1—figure supplement 2A and C*), which led to rescue of laminin binding capacity (*Figure 1—figure supplement 2D*).

### Ribitol and ribose rescue α-DG functional glycosylation in FP4 iPS cell-derived myotubes

Having developed this platform, we tested whether ribitol, a precursor for CDP-ribitol (*Figure 2A*), would be able to increase functional α-DG glycosylation in the human context using the FP4 patient-specific iPS cell-derived model. At the onset of terminal myogenic differentiation, we treated myogenic progenitors with increasing concentrations of ribitol (25 mM, 50 mM, 100 mM, and 200 mM) for 5 days. Following evaluation of cell morphology and IIH6 immunoreactivity (*Figure 2—figure supplement 1A and B*), the 50 mM concentration was chosen for the studies described here. Treated cells retained differentiation capacity, as shown by MHC levels (*Figure 2B*, *Figure 2—figure supplement 1C*), and exhibited rescue of α-DG functional glycosylation (*Figure 2B,C and D*). Importantly, this increase in functional glycosylation of α-DG was sufficient to increase binding between α-DG and laminin, as shown by the detection of laminin only in ribitol-treated FP4 myotubes (*Figure 2C*).

Since ribitol is endogenously generated by the reduction of ribose via an oxidoreductase (*Figure 2A*), we hypothesized that supplementing FP4 myogenic cells with ribose might also recover α-DG functional glycosylation. As before, we treated FP4 cultures with increasing concentrations of ribose, ranging from 5 to 100 mM. Because the lowest concentration of ribose able to enhance α-DG functional glycosylation was 10 mM and higher concentrations led to cell death (>50 mM), we chose the concentration of 10 mM for further analysis (*Figure 2—figure supplement 1D* to F). As shown in *Figure 2E,F and G*, we found a significant increase in functional glycosylation of α-DG upon 10 mM ribose supplementation. This increase was sufficient to enhance laminin binding capacity in FP4 myotubes (*Figure 2F*).

### Ribitol and ribose treatment is associated with significant increases in ribitol-5-P and CDP-ribitol

To determine the effect of ribitol and ribose supplementation on the synthesis of ribitol-5-P and CDP-ribitol, we quantified the levels of these PPP metabolites after 5 days of treatment by liquid chromatography with tandem mass spectrometry (LC/MS-MS). Quantification of each metabolite was determined based on generated standard curves (*Figure 3—figure supplement 1*). The data from this analysis revealed that both ribitol and ribose supplementation result in significant increases in ribitol, ribose, ribitol-5-P, and CDP-ribitol compared to untreated cultures (*Figure 3A*).

The reduction of ribose to ribitol has been suggested to be mediated by the sorbinil sensitive aldose reductase (AKR1B1) (*Gerin et al., 2016*). To determine whether inhibition of this aldose reductase would diminish α-DG functional glycosylation and counteract the ribose-mediated rescue, we treated WT cells at the onset of terminal differentiation with sorbinil. This resulted in a 40% reduction in IIH6 levels in WT myotubes (*Figure 3B and C*). Most importantly, sorbinil treatment counteracted the positive effect of ribose in FP4 myotubes by 63%, as shown by the diminished rescue of IIH6 levels (*Figure 3D and E*).

### Rescue of α-DG function by PPP metabolites is significantly enhanced by addition of NAD+

Since NAD+ has been shown to improve muscle function in the FKRP dystroglycanopathy zebrafish model (*Bailey et al., 2019*), we tested the effect of NAD+ alone or in combination with ribitol or ribose. We treated FP4 cultures with 100 μM of NAD+, as this concentration has been previously documented (*Goody et al., 2012*). In FP4 myotubes treated with NAD+ alone, we observed a small, yet significant increase in functional glycosylation of α-DG (*Figure 4—figure supplement 1*). However, when we combined NAD+ supplementation with PPP metabolites, we observed, on average, a 59% increase in IIH6 positivity in ribitol/NAD+ when compared to ribitol alone (*Figure 4A and C*). The same synergistic effect was observed when FP4 cells were treated with the NAD+/ribose combination, as levels of IIH6 increased on average 50% compared to ribose alone (*Figure 4B and D*). Importantly, in both cases, the combination also promoted increased laminin binding capacity (*Figure 4A and B*). We also tested the effect of ribitol and ribose in combination with NAD+ in

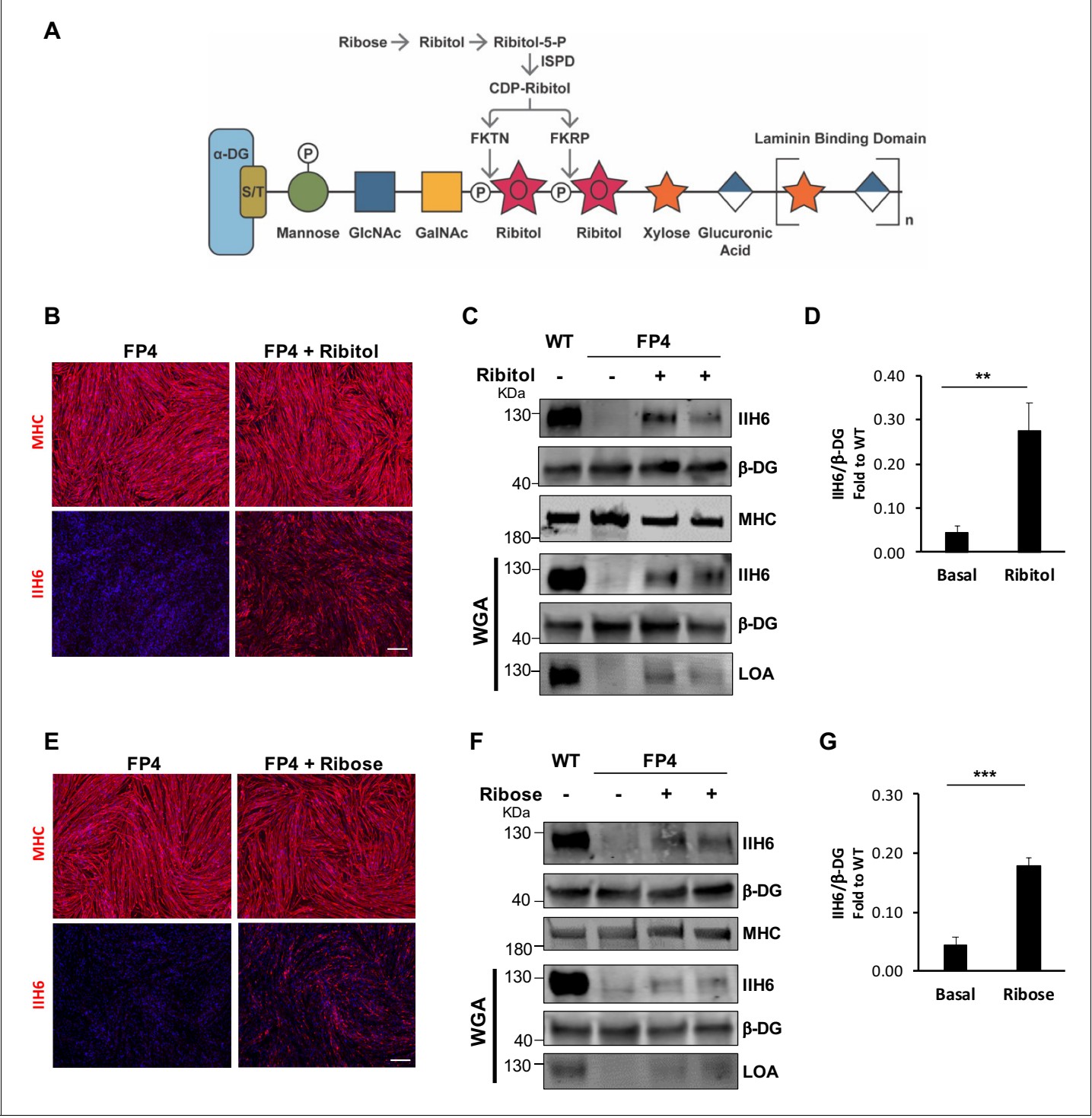

**Figure 2.** Ribitol and ribose rescue α-dystroglycan (α-DG) functional glycosylation in patient-specific Walker-Warburg syndrome (WWS) myotubes. (**A**) Scheme denoting the role of ribose and ribitol in the generation of Rbo5P, which is required by fukutin (FKTN) and fukutin-related protein (FKRP) for glycosylation of α-DG. (**B**) Representative immunostaining for myosin heavy chain (MHC) and IIH6 (in red) in WWS FP4 induced pluripotent stem (iPS) cell-derived myotubes that had been treated or not with ribitol. DAPI stains nuclei (in blue). Scale bar, 200 μm. (**C–D**) Western blot shows increased IIH6 staining in WWS FP4 myotubes upon ribitol supplementation. MHC (MF-20) and β-DG were used as differentiation and loading controls, respectively. Lower panel shows wheat germ agglutinin (WGA) pull-down for these samples, and laminin overlay assay (LOA) of WGA elutes shows laminin detection in ribitol-treated FP4 myotubes. Wild type (WT) myotubes were used as positive control. (**D**) Bar graph shows quantification of IIH6 (from C) normalized to β-DG and shown as the fold difference of WT. Error bars represent standard errors of five independent experiments. (**E**) Representative

*Figure 2 continued on next page*

*Figure 2 continued*

immunostaining for MHC and IIH6 (in red) in FP4 iPS cell-derived myotubes that had been treated or not with ribose. DAPI stains nuclei (in blue). Scale bar, 200 μm. (**F–G**) Western blot for IIH6 in FP4 myotubes that had been treated with ribose. MF-20 and β-DG were used as differentiation and loading controls, respectively. Lower panel shows representative WGA, and LOA shows laminin detection in ribose-treated FP4 myotubes. WT myotubes were used as positive control. (**G**) Bar graph shows increased IIH6 in ribose-treated myotubes. Quantification of IIH6 (**F**) was normalized to β-DG and shown as the fold difference of WT. Error bars represent standard errors of five independent experiments. Significance was evaluated by the unpaired Student's t test. *p<0.05, ***p<0.001.

The online version of this article includes the following figure supplement(s) for figure 2:

**Figure supplement 1.** Ribitol and ribose dose-response studies in induced pluripotent stem (iPS) cell-derived myotubes.

established myotubes. For this, we differentiated FP4 myogenic progenitors into myotubes, and 4 days later, added the compounds for 24–72 hr. Again, a synergistic effect was observed upon the combination of NAD+ with PPP metabolites. Ribitol/NAD+ treatment led to an 85% increase in IIH6 immunoreactivity compared to ribitol alone (*Figure 4E and F*). Likewise, ribose/NAD+ treatment on average doubled functional glycosylation of α-DG at 48 hr when compared to ribose alone (*Figure 4G and H*). These results support the beneficial effect of combining ribitol or ribose with NAD+ to enhance α-DG functional glycosylation.

Whereas ribitol and ribose rescue functional glycosylation of α-DG by increasing the generation of the FKRP substrate CDP-ribitol, the mechanism for NAD+ remains unclear. Since several forms of NAD act as cofactors for oxidoreductases, we hypothesized that NAD+ could enhance the generation of ribitol-5-P and CDP-ribitol. To test this, we quantified the levels of PPP metabolites in NAD+-treated cells by LC/MS-MS, as described above for ribitol/ribose (*Figure 3A*). We found that NAD+ treatment led to a small increase in ribose levels compared to untreated counterparts, but no significant differences were detected in ribitol-5-P and CDP-ribitol, as shown by comparing untreated vs. NAD+, ribitol vs. ribitol/NAD+, and ribose vs. ribose/NAD+ (*Figure 4—figure supplement 2*). These results suggest that the synergistic effect of NAD+ is independent of the FKRP substrate CDP-ribitol.

## Functional FKRP is indispensable for the rescue of α-DG functional glycosylation by ribitol and ribose

Because the null mutation for FKRP is embryonic lethal (*Chan et al., 2010*), most FKRP mutations are thought to have some residual activity. To determine whether such residual activity of FKRP is required for the effects observed upon ribitol and ribose supplementation, we generated an FKRP-deficient pluripotent stem cell line (FKRP knockout [KO]) using CRISPR/Cas9 genome editing. Immunostaining for MHC showed similar differentiation between FKRP KO myotubes and respective control WT counterparts (*Figure 5A*), but as anticipated, FKRP KO myotubes lacked functional glycosylation of α-DG, as evidenced by immunostaining and western blot to IIH6 (*Figure 5A and B*) and absence of laminin binding (*Figure 5B*). Importantly, treatment of FKRP KO myotubes with ribitol, ribose, or NAD+ did not rescue α-DG functional glycosylation (*Figure 5C*).

To investigate whether other mutations associated with the WWS phenotype are amenable to rescue by these metabolites, we introduced the WWS-clinically associated FKRP-C318Y mutation (*Beltran-Valero de Bernabé et al., 2004*) located in the zinc finger loop of the FKRP catalytic domain into WT iPS cells using CRISPR-Cas9 genome editing. Isogenic myotubes generated from FKRP-C318Y iPS cells displayed a similar phenotype to patient-specific FP4 myotubes (*Figure 5D*), thus confirming the in vitro WWS phenotype. We tested ribitol, ribose, or combinations with NAD+ supplementation in cultures of FKRP-C318Y myotubes, as described above for FP4, and none of the metabolites were able to rescue α-DG functional glycosylation in these cells (*Figure 5E*), suggesting that rescue is mutation specific.

To understand how WWS-associated mutations (*Figure 6A*) may interfere with the FKRP enzymatic activity, we turned to the recently deciphered crystal structure of FKRP (*Figure 6B*; *Kuwabara et al., 2020*). The zinc finger loop in the FKRP catalytic domain consists of four conserved cysteine residues (C289, C296, C317, C318) required for $Zn^{2+}$ ion binding. The C318Y mutation disrupts direct chelation to the $Zn^{2+}$ ion and sterically hinders the binding site from occupancy. This result suggests that this mutation will lead to significant, if not complete, loss of

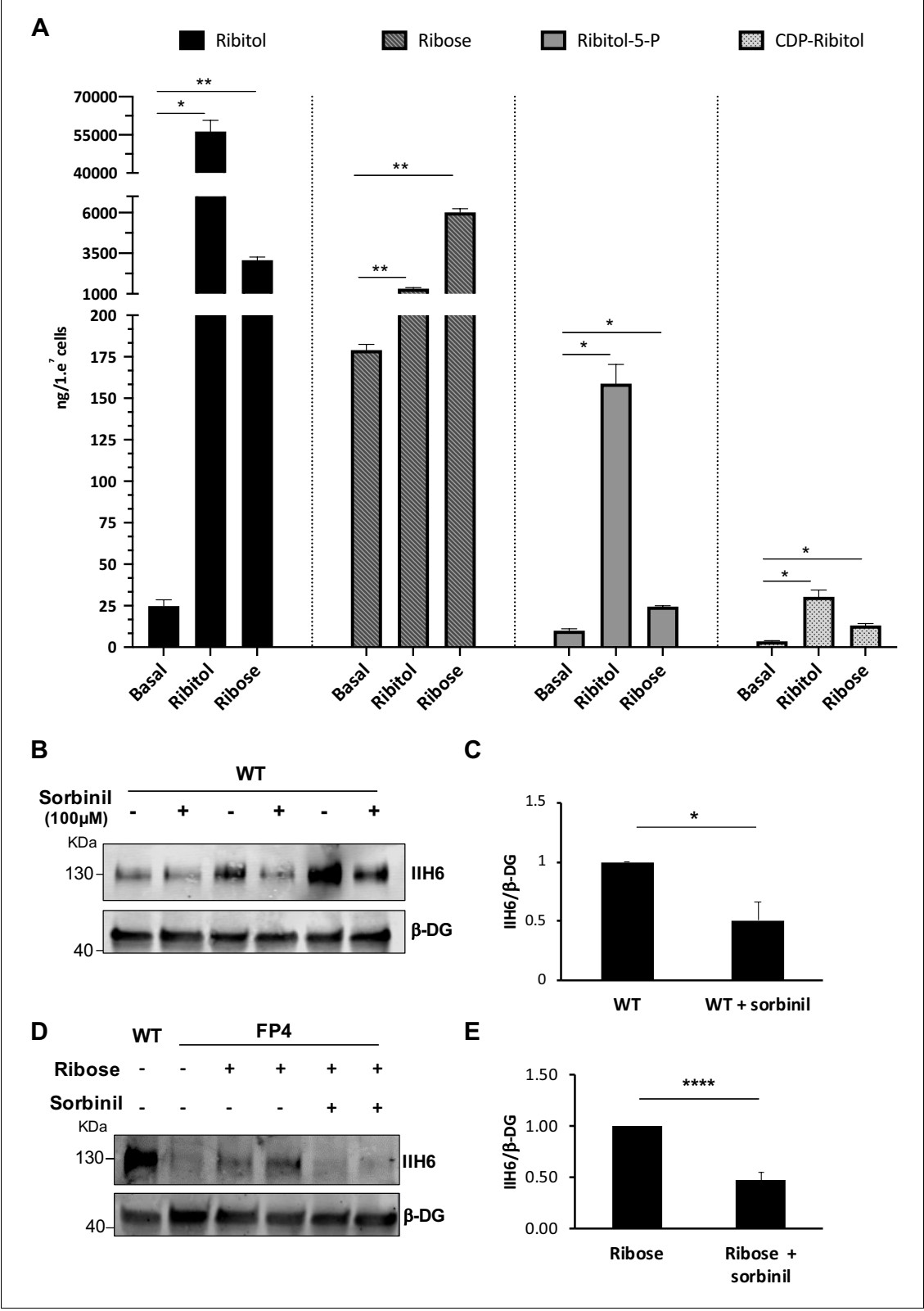

**Figure 3.** Ribitol and ribose increase the levels of ribitol-5-P and cytidine diphosphate (CDP)-ribitol in patient-specific Walker-Warburg syndrome (WWS) myotubes. (A) Detection of ribitol, ribose, ribitol-5-P, and CDP-ribitol in WWS FP4 myotubes that had been treated or not with ribitol or ribose for 5 days (n = 3 for each cohort). Error bars represent standard errors of three independent experiments. (B–C) Sorbinil treatment for 5 days results in decreased α-dystroglycan (α-DG) functional glycosylation, as shown by western blot for IIH6 in wild type (WT) myotubes (B). β-DG was used as
*Figure 3 continued on next page*

*Figure 3 continued*

loading control. (C) Bar graph shows quantification of IIH6 (from B) normalized to β-DG and shown as the fold difference of WT. Error bars represent standard errors of three independent experiments. (D–E) Sorbinil treatment counteracts the positive effect of ribose on α-DG functional glycosylation. (D) Representative western blot shows reduction of IIH6 staining in FP4 myotubes that had been treated with both ribose and sorbinil. β-DG was used as loading control. (E) Bar graph shows quantification of IIH6 (from D) normalized to β-DG and shown as the fold difference of FP4 + ribose. Error bars represent standard errors of five independent experiments. Significance was evaluated by the one-way ANOVA followed by the Sidak's multiple comparison test in (A) and the unpaired Student's t test in (C and E). *$p<0.05$, **$p<0.01$, ***$p<0.001$.
The online version of this article includes the following figure supplement(s) for figure 3:

**Figure supplement 1.** Liquid chromatography with tandem mass spectrometry (LC/MS-MS) standard curves.

enzymatic function, and therefore metabolite-mediated rescue is not possible. On the other hand, F473 makes up a small hydrophobic pocket with L348, I357, W359, V477, and P481 that is essential for CDP-ribitol substrate binding within the catalytic domain. The mutation of F473 to cysteine (F473C) present in the FP4 patient sample leads to a free energy change in substrate binding affinity of +4.3 kcal/mol. This difference suggests that the F473C mutation results in destabilization of the Michaelis complex formation with diminished enzyme efficiency, and therefore, metabolite supplementation that increases CDP-ribitol levels allows for increased FKRP activity. These results indicate that functional FKRP is required for rescue of α-DG functional glycosylation by ribitol, ribose, and NAD+.

## Discussion

Myoblasts harvested from patients are commonly used to model muscular dystrophies in vitro. However, in cases like WWS, the short lifespan along with the difficulty in obtaining tissue from patients represents major hurdles in establishing patient-specific myoblasts lines. The generation of patient-specific iPS cells circumvents the restricted patient tissue availability and the limited cell proliferation capacity seen in ex vivo expanded primary cells (*Kondo et al., 2013*; *McKeithan et al., 2020*; *Sampaziotis et al., 2015*; *Young et al., 2016*).

To date, several experimental studies in animal models have provided evidence supporting the potential therapeutic application of gene therapy (*Gicquel et al., 2017*; *Xu et al., 2013*) and cell therapy (*Azzag et al., 2020*; *Frattini et al., 2017*) for FKRP-associated muscular dystrophies, but these studies are still at early stages, and therefore, currently there are no clinical trials underway. In this study, we show that PPP metabolites are able to increase functional glycosylation of α-DG in WWS patient-specific iPS cell-derived myotubes associated with FKRP mutations (FP4). Besides ribitol, we show that ribose is also able to provide significant increase in IIH6 immunoreactivity in FKRP mutants, which is accompanied by rescue of laminin binding. Our results indicate that both these PPP metabolites increase ribitol-5-P and CDP-ribitol levels in FP4-treated myotubes. The enhanced functional glycosylation of α-DG in FP4 mutant myotubes is hypothesized to be due to increased CDP-ribitol levels leading to increased ribitol-5-P transferase activity in the disease-causing FKRP mutant.

We show for the first time that NAD+ can increase functional glycosylation of α-DG in a human WWS FKRP model, and when combined with ribitol or ribose, can significantly potentiate the rescue of the muscle pathology in vitro. Studies in dystroglycan (*dag1*) and FKRP zebrafish mutants have demonstrated a beneficial effect for NAD+ (*Bailey et al., 2019*; *Goody et al., 2012*). Although the mechanism is not entirely elucidated, NAD+ was reported to promote increased ADP-ribosylation of integrin receptors, which in turn increase integrin and laminin binding, increase laminin-111 organization and subcellular localization of paxillin to cell adhesion complexes (*Goody et al., 2012*; *Zolkiewska, 2005*). Interestingly, overexpression of paxillin rescues muscle structure in *dag1* but not in FKRP mutants, suggesting a different mechanism of action (*Bailey et al., 2019*). Previous studies in two Duchenne MD mouse models showed that NAD+ improved muscle function via reduced parylation, as well as increased mitochondria function and expression of structural proteins (*Ryu et al., 2016*). Our data on the quantification of core α-DG in FKRP-C318Y myotubes that had been treated or not with NAD+ revealed a significant increase in core α-DG upon NAD+ treatment, whereas β-DG levels remained unchanged (*Figure 5—figure supplement 1*), suggesting that NAD+ supplementation

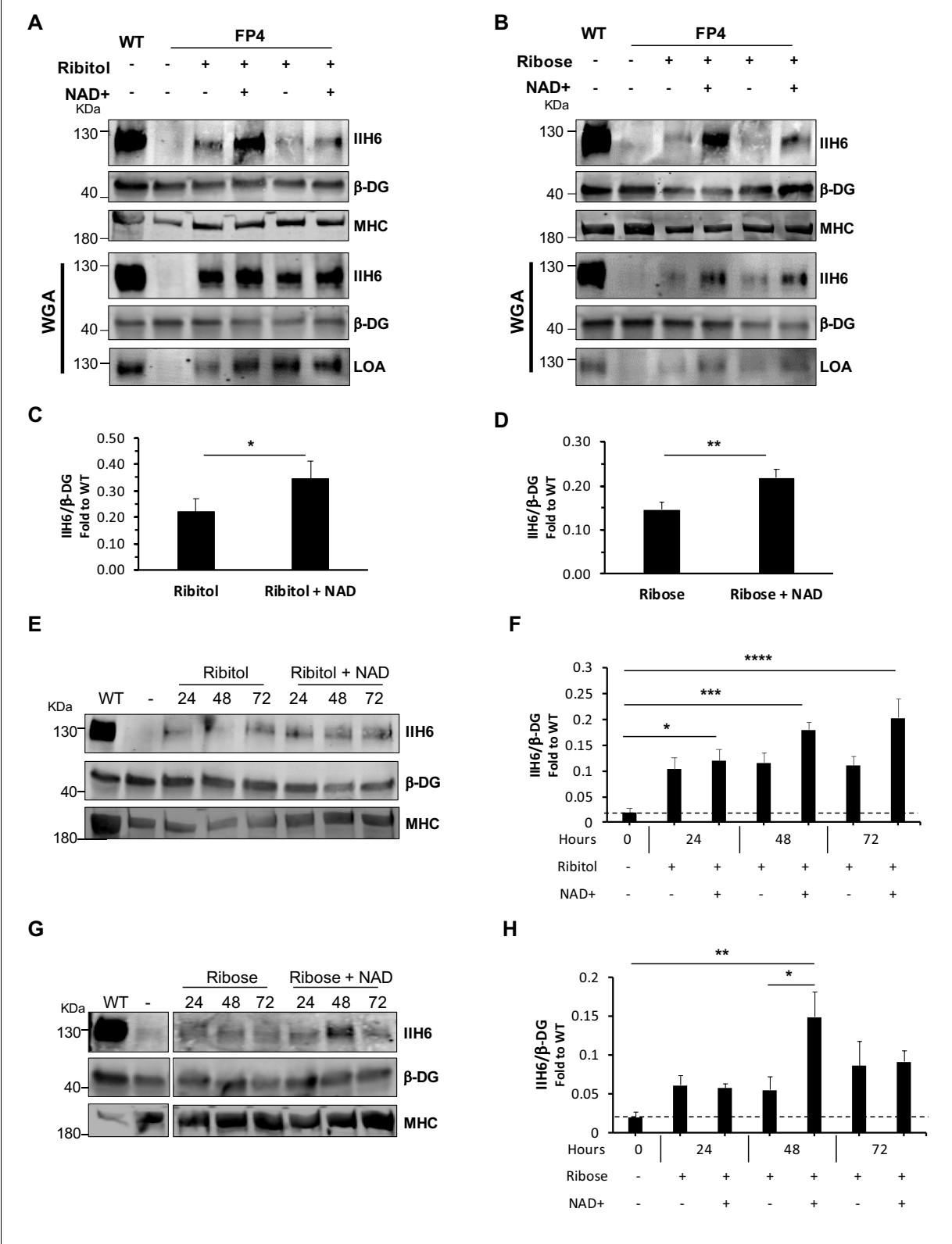

**Figure 4.** NAD+ potentiates the capacity of ribitol and ribose to rescue α-dystroglycan (α-DG) functional glycosylation. (A–B) Western blots show increased IIH6 staining in Walker-Warburg syndrome (WWS) FP4 myotubes that had been supplemented with ribitol and NAD+ compared to ribitol alone (A) or with ribose and NAD+ compared to ribose alone (B). Myosin heavy chain (MHC) (MF-20) and β-DG were used as differentiation and loading controls, respectively. Lower panel shows wheat germ agglutinin (WGA) pull-down for these samples, and respective laminin overlay

*Figure 4 continued on next page*

*Figure 4 continued*

assay (LOA) of elutes shows increased laminin detection in FP4 myotubes that were treated with ribitol/NAD+ or ribose/NAD+. Wild type (WT) myotubes were used as positive control. (**C–D**) Bar graph shows quantification of IIH6 (A and B, respectively) normalized to β-DG and shown as the fold difference of WT. Error bars represent standard errors of 8 (for C) or 7 (for D) independent experiments. (**E–H**) Western blot of IIH6 in FP4 D4 differentiated myotubes treated with ribitol and ribitol/NAD+ (**E**) or ribose and ribose/NAD+ (**F**) for 24, 48, or 72 hr. MF-20 and β-DG were used as differentiation and loading controls, respectively. (**G–H**) Bar graph shows quantification of IIH6 (from E and F, respectively) normalized to β-DG and shown as the fold difference of WT. Error bars represent standard errors of four independent experiments. Significance was evaluated by the paired Student's t test in (**C and D**) and by the one-way ANOVA followed by the Sidak's multiple comparison test in (**F and H**). *p<0.05, **p<0.01, ***p<0.001, ****p<0.0001.

The online version of this article includes the following figure supplement(s) for figure 4:

**Figure supplement 1.** NAD+ rescue functional glycosylation of α-dystroglycan (α-DG) in FP4 induced pluripotent stem (iPS) cell-derived myotubes.

**Figure supplement 2.** NAD+, ribitol/NAD+, and ribose/NAD+ do not increase ribitol-5-P or cytidine diphosphate (CDP)-ribitol in FP4 myotubes.

specifically increases α-DG. Although further studies are required to elucidate the mechanism by which NAD+ may lead to increased α-DG, a plausible hypothesis is a post-translational effect.

Importantly, we show that functional FKRP mediates the rescue by ribitol, ribose, and NAD+ since FKRP KO and FKRP-C318CY myotubes do not show IIH6 rescue upon treatment with any of these compounds or combinations. Based on the recently reported FKRP crystal structure (*Kuwabara et al., 2020*), C318 is located in the zinc finger loop (G288 to C318) of the FKRP catalytic domain, which has been proposed to be of fundamental importance for the catalytic activity of FKRP (*Kuwabara et al., 2020*). Our results suggest that the ability of PPP metabolites to partially rescue α-DG functional glycosylation is mutation dependent. Although further studies are required to determine which patients could benefit from this potential treatment, our results suggest that phenotypes associated with mutations in the zinc finger loop region may not be rescued by ribitol and ribose, whereas FKRP mutations in other regions of the catalytic domain are amenable to rescue, as shown for FP4. This is in line with previous studies in ISPD fibroblasts, in which rescue of functional glycosylation of α-DG was found to be mutation dependent (*Gerin et al., 2016*; *van Tol et al., 2019*).

Dietary interventions can provide a feasible and economically accessible solution for the treatment of MD associated with CDP-ribitol defects. Although ribitol/NAD+ showed promising results in our model, clinical trials to assess the safety of ribitol are still necessary. On the other hand, ribose is a commercially available supplement, and to date, with no major side effects in humans (*Dodd et al., 2004*; *Seifert et al., 2017*; *Thompson et al., 2014*). Furthermore, NAD+ levels can be increased by several vitamin B3 forms, such as nicotinic acid (niacin) and nicotinamide riboside, which have been investigated, showing no major side effects (*Elhassan et al., 2019*; *Guyton and Bays, 2007*; *Pirinen et al., 2020*). Although future research studies are necessary to determine the optimal dosage of the combined approach, the safety record of these compounds justifies using ribose/NAD + as potential candidates to treat FKRP-associated MD. Together, our results support the use of iPS cell-derived myotubes as a reliable platform for in vitro disease modeling and drug screening. Importantly, our data provide a rationale for the potential use of ribitol/NAD+ and ribose/NAD+ as therapeutics to increase α-DG functional glycosylation in patients with FKRP mutations.

# Materials and methods

## Key resources table

| Reagent type (species) or resource | Designation | Source or reference | Identifiers | Additional information |
|---|---|---|---|---|
| Cell line (*Homo sapiens*, male) | FP4 | This study | | Available from the Anne Bang lab |
| Genetic reagent (*Homo sapiens*, male) | FKRP C318Y | This study | TC1133 FKRP C318Y | Available from the Rita Perlingeiro lab |
| Genetic reagent (*Homo sapiens*, male) | FKRP KO | This study | H9 FKRP KO | Available from the Rita Perlingeiro lab |

*Continued on next page*

*Continued*

| Reagent type (species) or resource | Designation | Source or reference | Identifiers | Additional information |
|---|---|---|---|---|
| Cell line (*Homo sapiens*, male) | WT | PMID:22560081 | PLZ | Control line, available from the Rita Perlingeiro lab |
| Cell line (*Homo sapiens*, male) | Parental WT FKRP KO | WiCell | H9 | ESC control line (WA09) |
| Cell line (*Homo sapiens*, male) | Parental WT FKRP C318Y | PMID:26411904 | TC-1133 | Control line, available with RUCDR Infinite Biologics |
| Chemical compound, drug | CHIR99021 | Tocris | Cat# 4423 | 10 µM |
| Chemical compound, drug | LDN193189 | Cayman chemical | Cat# 19396 | 200 nM |
| Chemical compound, drug | SB431542 | Cayman chemical | Cat# 13031 | 10 µM |
| Chemical compound, drug | DAPT | Cayman chemical | Cat# 13197 | 10 µM |
| Chemical compound, drug | Dexamethasone | Cayman chemical | Cat# 11015 | 10 µM |
| Chemical compound, drug | Forskolin | Cayman chemical | Cat# 11018 | 10 µM |
| Chemical compound, drug | Ribitol | Sigma-Aldrich | Cat# A5502 | 50 mM |
| Chemical compound, drug | D-(−)-ribose | Sigma-Aldrich | Cat# R9629 | 10 mM |
| Chemical compound, drug | NAD+ | Sigma-Aldrich | Cat# N0632 | 100 µM |
| Chemical compound, drug | Sorbinil | Sigma-Aldrich | Cat# S7701 | 100 µM |
| Chemical compound, drug | Doxycycline | Sigma-Aldrich | Cat# D9891 | 1 µg/ml |
| Recombinant protein | Recombinant human FGF-basic | Peprotech | Cat# 100-18B | 5 ng/ml |
| Antibody | Anti-alpha dystroglycan (mouse monoclonal) | Millipore | Cat# 05–593, RRID:AB_309828 | Dilution 1:1000 (WB) |
| Antibody | Anti-alpha dystroglycan (mouse monoclonal) | DSHB | Cat# IIH6 C4, RRID:AB_2617216 | Dilution 1:50 (IF) |
| Antibody | Anti-human dystroglycan (sheep polyclonal) | R and D Systems | Cat# AF6868, RRID:AB_10891298 | Dilution 1:1000 (WB) |
| Antibody | Anti-MHC (mouse monoclonal) | DSHB | Cat# MF20, RRID: AB_2147781 | Dilution 1:50 (IF) 1:200 (WB) |
| Antibody | Anti-laminin (rabbit polyclonal) | Sigma-Aldrich | Cat# L9393, RRID:AB_477163 | Dilution 1:1000 (WB) |
| Antibody | Anti-beta dystroglycan, concentrated (mouse monoclonal) | DSHB | Cat# MANDAG2 clone 7D11, RRID:AB_2211772 | Dilution 1:1500 (WB) |
| Antibody | Anti-OCT3/4 (mouse monoclonal) | SCBT | Cat# C-10, RRID: AB_628051 | Dilution 1:50 (IF) |
| Antibody | Anti-SOX2 (goat polyclonal) | SCBT | Cat# Y-17, RRID: AB_2286684 | Dilution 1:50 (IF) |
| Antibody | Anti-NANOG (mouse monoclonal) | SCBT | Cat# H-2, RRID: AB_10918255 | Dilution 1:50 (IF) |
| Antibody | Anti-SSEA4 (mouse monoclonal) | SCBT | Cat# sc-21704, RRID: AB_628289 | Dilution 1:50 (IF) |

*Continued on next page*

*Continued*

| Reagent type (species) or resource | Designation | Source or reference | Identifiers | Additional information |
|---|---|---|---|---|
| Antibody | Alexa fluor 555 goat anti-mouse IgG and IgM (goat polyclonal) | Thermo Fisher Scientific | Cat# A-21424, RRID: AB_141780 Cat# A-21042, RRID:AB_2535711 | Dilution 1:500 (IF) |
| Antibody | DyLight 680 anti-rabbit IgG (goat polyclonal) | Thermo Fisher Scientific | Cat# 35568, RRID:AB_614946 | Dilution 1:10000 (WB) |
| Antibody | DyLight 680 anti-mouse IgM (goat polyclonal) | Thermo Fisher Scientific | Cat# SA5-10154, RRID:AB_2556734 | Dilution 1:10000 (WB) |
| Antibody | DyLight 680 anti-sheep IgG (rabbit polyclonal) | Thermo Fisher Scientific | Cat# SA5-10058, RRID:AB_2556638 | Dilution 1:10000 (WB) |
| Antibody | DyLight 800 anti-mouse IgG (goat polyclonal) | Thermo Fisher Scientific | Cat# SA5-10176, RRID:AB_2556756 | Dilution 1:10000 (WB) |
| Other | DAPI stain | SCBT | sc-3598 | (1.5 µg/ml) |
| Other | Laminin from Engelbreth-Holm-Swarm murine sarcoma basement membrane | Millipore Sigma | Cat# L2020 | 1 µg/µl |

## iPS cell reprogramming and cell lines

FKRP mutant fibroblasts obtained from a 1-year-old male patient (*Kava et al., 2013*) were reprogrammed into iPS cells, named FP4, using the CytoTune-iPS 2.0 Sendai Reprogramming Kit (Thermo Fisher Scientific) using feeder-free conditions, according to the manufacturer's instructions. FP4 iPS cells were passaged with ReLeSR (STEMCELL Technologies) and cultured on Matrigel-coated dishes using mTeSR1 medium (STEMCELL Technologies). Newly generated and previously described WT iPS/embryonic stem (ES) cells (*Darabi et al., 2012*; *Selvaraj et al., 2019b*) are listed in the key resources table. Cell lines were authenticated by verification of genetic mutation by sanger sequencing. All cell lines were negative for mycoplasma contamination.

## Mice and teratoma studies

Experiments were carried out according to protocols (protocol ID 2002-37833A) approved by the University of Minnesota Institutional Animal Care and Use Committee. NOD scid gamma (NSG) mice (Jackson laboratory) were used to perform teratoma studies. FP4 cells ($1.5 \times 10^6$) were suspended in a 1:1 Dulbecco's Modified Eagle Medium: Nutrient Mixture F-12 (DMEM-F12, ThermoScientific) and Matrigel (Corning) solution and injected in the quadriceps of NSG mice. The teratoma was harvested 2 months after injection.

## Myogenic differentiation and cell culture

Inducible PAX7 FP4 human iPS cells were generated by lentiviral transduction of the pSAM2-PAX7-IRES-GFP and pFUGW-rtTA constructs. Inducible Pax7 WT iPS cells were generated previously (*Darabi et al., 2012*), and they were maintained on Matrigel-coated flasks using mTeSR 1 (STEMCELL Technologies). iPAX7-iPS cells were dissociated with Accumax (Innovative Cell Technologies), and $1 \times 10^6$ cells were plated on a 6 cm non-adherent Petri dishes using mTeSR1 medium supplemented with 10 µM Y-27632 (ROCK inhibitor) and incubated on a shaker at 60 rpm (day 0). On day 2, the medium was replaced with embryoid body (EB) differentiation medium (15% fetal bovine serum (FBS), 10% horse serum, 1% KnockOut Serum Replacement, 1% GlutaMax, 1% penicillin-streptomycin, 50 µg/ml ascorbic acid, and 4.5 mM monothioglycerol in Iscove's modified Dulbecco's medium) supplemented with 10 µM CHIR990217 (GSK3 inhibitor). After 2 days of incubation of EBs in suspension, the medium was replaced with fresh EB differentiation medium containing 10 µM SB-431542 and 200 nM LDN-193189. On day 5, 1 µg/ml doxycycline was added to promote PAX7 induction. After 24 hr the media was changed with fresh (EB) differentiation medium with 1 µg/ml doxycycline. Day 8 EBs were collected and plated as a monolayer on gelatin-coated flasks using EB differentiation medium supplemented with 10 ng/ml human basic fibroblast growth factor and 1 µg/ml doxycycline. On day 12, GFP+ cells (PAX7+ myogenic progenitors) were sorted using a FACS

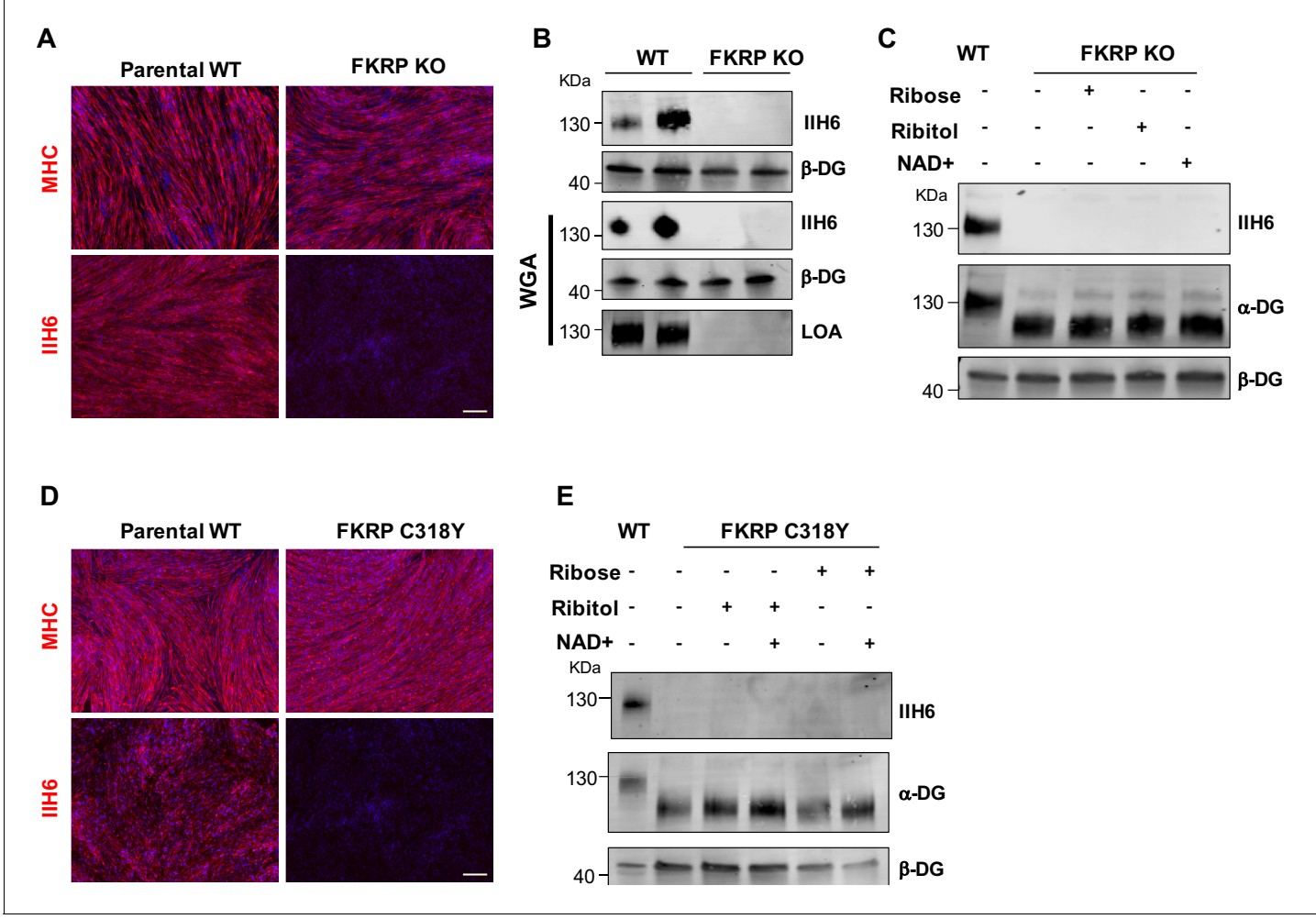

**Figure 5.** Functional fukutin-related protein (FKRP) is required for NAD+ ribitol/ribose-mediated rescue of α-dystroglycan (α-DG) functional glycosylation. (**A**) Representative immunostaining of FKRP knockout (KO) and respective control wild type (WT) isogenic PS cell-derived myotubes for myosin heavy chain (MHC) and IIH6 (in red). DAPI stained nuclei (in blue). Scale bar, 200 μm. (**B**) Western blot for IIH6 shows absence of α-DG functional glycosylation in FKRP KO myotubes. β-DG was used as loading control. Lower panel shows representative wheat germ agglutinin (WGA) pull-down for these samples and laminin overlay assay (LOA) shows the lack of laminin detection in FKRP KO myotubes. Parental WT myotubes were used as positive control. (**C**) Deletion of FKRP abolishes ribitol, ribose, and NAD+ mediated rescue of α-DG functional glycosylation, as shown by western blot for IIH6. β-DG was used as loading control. (**D**) Representative immunostaining of parental WT (WT-2) and FKRP C318Y iPS cell-derived myotubes for MHC (upper panel) and IIH6 (lower panel). DAPI stains nuclei (in blue). Scale bar, 200 μm. (**E**) FKRP C318Y abolishes ribitol, ribose, and combinations with NAD+ mediated rescue of α-DG functional glycosylation, as shown by western blot for IIH6. β-DG was used as loading control.
The online version of this article includes the following figure supplement(s) for figure 5:

**Figure supplement 1.** NAD+ treatment in FKRP C318Y myotubes increases α-dystroglycan (α-DG) levels.

Aria II (BD Biosciences) and expanded on gelatin-coated flasks using the same medium. At 90% cell density, cells were passaged using Trypsin-EDTA (Gibco) and replated on new gelatin-coated flasks.

Myogenic progenitors were terminally differentiated into myotubes by growing them to confluency and then switching to terminal differentiation medium, which consisted of DMEM low glucose supplemented with 2% horse serum, 1% insulin-transferrin-selenium, 1% penicillin-streptomycin, 10 μM SB-431542, 10 μM LY-374973, 10 μM Forskolin, and 10 μM dexamethasone (*Selvaraj et al., 2019b*). At this point, cultures were exposed to different treatments as follows: ribitol (A5502, Sigma-Aldrich), D-(−)-ribose (R9629, Sigma-Aldrich), 100 μM NAD+ (N0632, Sigma-Aldrich), and/or 100 μM sorbinil (S7701, Sigma-Aldrich). Media was replenished on day 3 of differentiation, and myotubes characterization was performed after 5–8 days of terminal differentiation.

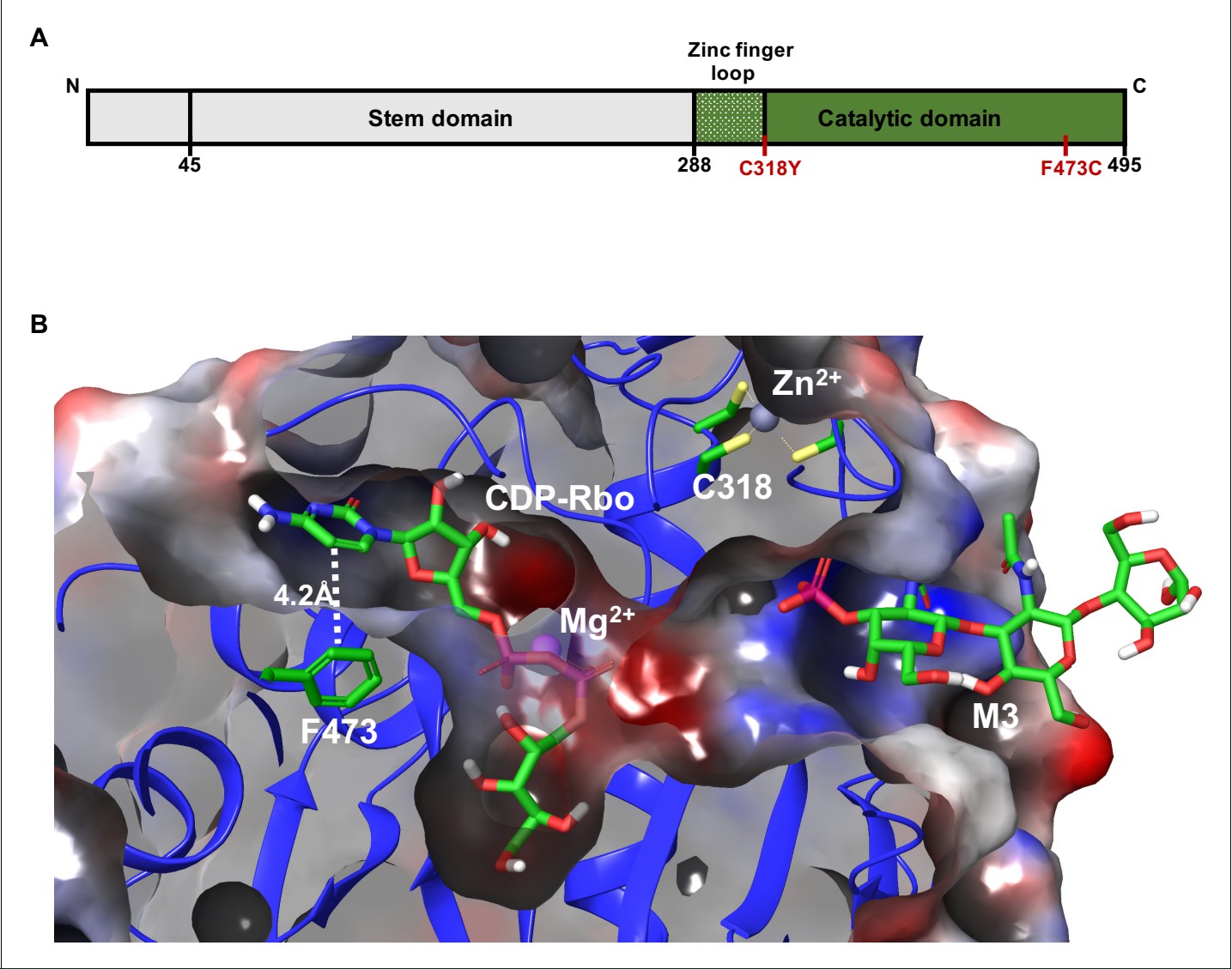

**Figure 6.** Model depicting the Walker-Warburg syndrome (WWS)-associated mutations in the fukutin-related protein (FKRP) catalytic domain. (**A**) Schematic representation of FKRP protein highlighting the N-terminal stem domain and the localization of C318Y and F473C mutations in the C-terminal catalytic domain. (**B**) Crystal structure of the catalytic domain of FKRP highlights the cytidine diphosphate-ribitol (CDP-Rbo) and α-DG M3 glycan binding sites within its catalytic domain. The FKRP C318Y mutation is one of the four required cysteine residues located in the zinc finger loop. The F473C mutation present in the FP4 patient-specific induced pluripotent stem (iPS) cell line is located adjacent to the pyrimidine ring of CDP-Rbo essential for substrate binding. An electrostatic potential surface highlights the overall cavity size and the electrostatic complementarity between the bound substrates and FKRP, with the neutral, negatively, and positively charged surfaces colored in the white, red, and blue, respectively. The model is based on the PDB:6KAM structure.

### Cytogenetic analysis

Live iPS cells were submitted to the Cytogenomics core at the University of Minnesota Masonic Cancer Center for G-band karyotype analysis. Cells were treated with colcemid for 3 hr to arrest cells, and 20 different metaphases were analyzed at a resolution of 400–450 band level.

### Generation of FKRP isogenic cell lines

To generate the FKRP KO pluripotent stem cell line, the previously published gRNA (CATGCGGCTCACCCGCTGCCAGG) targeting the start codon of FKRP (*Yagi et al., 2016*) was cloned into pSpCas9(BB)−2A-GFP (PX458; Addgene plasmid # 48138) (*Ran et al., 2013*). The ES cell line H9 was nucleofected using the Human Stem Cell Nucleofector Kit 1 (Lonza) and sorted for GFP at 48 hr

post-nucleofection. ES cells were expanded, and IIH6 negative cells were sorted by FACS. The deletion was confirmed by sequencing.

FKRP C318Y mutant iPS cells were generated using an HDR donor vector as previously described (*Selvaraj et al., 2019a*). FKRP exon 4 carrying the c.953 G>A (p.318 C>Y) mutation was cloned upstream of GFP-2A-neoR cassette (Dhoke et al., in prep). Gene editing was carried out using a ribonucleoprotein based delivery of guide RNA (Synthego) and Hifi Cas9 protein (IDT). Following antibiotic selection, FACS purified IIH6 negative cells were expanded and subjected to single cell cloning.

The FKRP vector was generated by cloning the full-length FKRP coding sequence from Dharmacon (clone 3160297) into pSAM-ires-mCherry vector (*Bosnakovski et al., 2008*). Plasmids were prepared using an Endofree Midiprep kit (Nucleobond). Lentiviruses were produced by co-transfection of the transfer vector and the packaging constructs (pVSV-G and pΔ8.74) into HEK 293 T cells. Transfections were performed using Lipofectamine LTX with Plus Reagent (Invitrogen) following manufacturer instructions. Supernatants containing the lentiviral particles were collected 36 hr after transfection and passed through a 0.45 µm filter. Myogenic progenitors were transduced with pSAM-ires-mCherry (empty-LV FP4) or pSAM-FKRP-iresmCherry (FKRP FP4), and subsequently mCherry-positive cells were purified by FACS.

## IIH6 FACS analysis

IIH6 staining for FACS was performed as previously described with minor modifications (*Rojek et al., 2007*). iPS cells were washed once with phosphate buffer saline (PBS) and then harvested using enzyme-free cell dissociation buffer (Gibco) following the manufacturer's instructions. Cells were collected, centrifuged, washed with PBS, and then resuspended in PBS supplemented with 10% FBS (PBSF) in the presence of Fc Block (1 µl/million cells – BD Bioscience) and incubated for 5 min. Staining was performed by adding 1 µl of anti-α-DG antibody IIH6C4 (Millipore) or normal mouse IgM (Santa Cruz Biotechnology) antibody per million cells followed by 20 min incubation on ice. Cells were then washed with PBS and labeled with 488- or 555-conjugated secondary antibodies (1:500 in FACS buffer) for 20 min on ice in the dark. Cells were washed with PBS and filtered through a 70 µm strain to remove cell clumps, then resuspended in PBSF. Samples were sorted using a FACS Aria II (BD Biosciences).

## Immunoblot analysis and WGA pull-downs

Frozen cells were homogenized in Tris-Buffer Saline (TBS, 50 mM Tris-Cl, pH 7.5, 150 mM NaCl) with 1% Triton X-100 and a cocktail of protease inhibitors (Complete – Millipore-Sigma) at 4°C by vortexing and then centrifuged for 30 min at 30000 g. Solubilized proteins from the supernatant were quantified with Bradford reagent (Millipore-Sigma). Protein samples were prepared in Laemmli Sample Buffer (LSB, BioRad). WGA pull-downs were performed using 350–600 µg of protein lysate that was loaded on 35–60 µl of WGA-bound agarose beads (Vector Laboratories, Inc) and incubated with end-over-end mixing at 4°C overnight. After three washes with PBS (150 mM NaCl, 8 mM NaH$_2$PO$_4$, 42 mM Na$_2$HPO$_4$, pH 7.5) with 0.1% Triton X-100, bound protein was eluted with 2x LSB and incubated at 100°C for 5 min. Protein samples were separated on 4–15% using precast polyacrylamide gel (BioRad) by electrophoresis and then transferred to Immobilon-FL PVDFmembranes (Millipore) for detection with the indicated antibodies using Licor's Odyssey Infrared Imaging System. Total protein detection using was preformed using LI-COR REVERT kit according to the manufacturer's instructions. Used antibodies are described in the key resources table.

## Laminin overlay assay

The LOA was performed as previously described with minor modifications (*Pall et al., 1996*). Briefly, 20 µl of WGA purified samples were separated on 4–15% SDS-polyacrylamide gels by electrophoresis and then transferred to Immobilon-FL PVDF membranes. Transfers were blocked with PBS and 5% nonfat dry milk for 1 hr at room temperature, and then briefly rinsed with TBS and incubated for 2 hr at room temperature in TBS containing 1 mM CaCl$_2$, 1 mM MgCl$_2$ (TBSS), 3% bovine serum albumin (BSA), and 1 mg/ml native laminin (L2020, Sigma). Transfers were washed twice for 10 min in TBSS and incubated overnight at 4°C with TBSS 3% BSA and anti-laminin (L9393, Sigma). Afterward, the membrane was washed with TBSS twice for 10 min and incubated with anti-rabbit DyLight 680 for 45 min at room temperature. Finally, membranes were washed with TBSS and visualized

using Licor's Odyssey Infrared Imaging System. As a negative control, TBSS without 1 mM $CaCl_2$ was used during incubation and washes.

## Metabolite extraction and LC/MS-MS analysis

Ribitol-5-phosphate and CDP-ribitol were synthesized by Z Biotech (Aurora, CO). Myogenic progenitors were serum-starved after changing to differentiation medium only or supplemented with ribitol, ribose, ribitol/NAD+, ribose/NAD+, or NAD+ for 5 days, washed with cold PBS three times and harvested by scrapping the cells. In a blinded manner, samples were subjected to the following procedures. Cells were homogenized with 300 µl of MeOH:acetonitrile (1:1) and then centrifugated for 5 min at 11,000 rpm. The supernatants were removed, transferred to individual wells of 96-well plate, and analyzed by LC/MS-MS. An Applied Biosystems Sciex 4000 (Applied Biosystems, Foster City, CA) equipped with a Shimadzu HPLC (Shimadzu Scientific Instruments, Inc, Columbia, MD) and Auto-sampler (LEAP Technologies, Carrboro, NC) were used to detect ribitol, ribose, ribitol-5-P, and CDP-ribitol. The analysis of metabolites was performed by Z Biotech as described previously (*Cataldi et al., 2018*).

## Molecular modeling

Modeling of FKRP with its CDP-ribitol and M3 substrates (PDB: 6KAM) (*Kuwabara et al., 2020*) was carried out using the Schrodinger modeling suite package (*Schrödinger Release 2018-4, 2018*). All missing side chains and hydrogens atoms were added according to the default protein preparation protocol at pH 7.0, followed by energy minimization using OPLS2005 force field (*Jorgensen et al., 1996*) to optimize all hydrogen-bonding networks. The crystallographic $Ba^{2+}$ ion was replaced by its native $Mg^{2+}$ ion. The relative change in substrate-binding free energy due to the effect of mutation, $\Delta\Delta G_{bind}$(F473C), was performed based on the molecular mechanics generalized Born solvent accessible method (*Still et al., 1990*). It is evaluated as the difference in the protein stability between the unbound and bound states of FKRP and its F473C mutant.

## Immunofluorescence analysis

Immunofluorescence staining was performed by fixing cells with 4% paraformaldehyde in PBS for 10 min at 4°C, followed by permeabilization with 0.1% Triton in PBS and blocking with 3% BSA in PBS, before incubation with the primary antibodies. Samples were rinsed with PBS, blocked with 3% BSA in PBS, and then incubated with DAPI and respective secondary antibodies. Antibodies used in this study are described in the key resources table.

## RT-qPCR

Samples were collected with TRIzol Reagent (Invitrogen), and RNA was purified using a Direct-zol RNA Miniprep Plus Kit (Zymo Research). Purified RNA was quantified with NanoDrop 2000 (Thermo Fisher Scientific) and retrotranscribed using SuperScript VILO cDNA Synthesis Kit (Invitrogen) following the manufacturer's instructions. Gene expression analyses were performed using the cDNA corresponding to 12.5 ng of starting RNA for each reaction. The RT-qPCR analysis was performed using TaqMan Universal PCR Master Mix and TaqMan probes (Applied Biosystems).

## Statistical analysis

For comparisons of two independent samples, we used the unpaired or paired Student's t test. For comparisons of multiple groups, we used the two-way ANOVA followed by the Tukey's multiple comparisons test. The one-way ANOVA followed by the Sidak's multiple comparisons test was used when measuring one variable. p-values < 0.05 were considered significant. Statistical comparisons were performed using GraphPad Prism software.

## Acknowledgements

This project was supported by NIH grants R01 AR071439 and AR055299 (RCRP), the LGMD2I Research Funds (RCRP and AGB). COC was supported by PINN MICITT Costa Rica. We are grateful to Jiri Vajsar for providing the WWS patient sample. We thank Lila Habib and James Kiley for their contribution in the generation of FP4 iPS cells and myogenic progenitors, respectively. The

monoclonal antibody to MHC and the IIH6 antibody were obtained from the Developmental Studies Hybridoma Bank developed under the auspices of the NICHD and maintained by the University of Iowa. We are thankful to Cynthia Faraday for graphical design.

## Additional information

### Funding

| Funder | Grant reference number | Author |
| --- | --- | --- |
| National Institute of Arthritis and Musculoskeletal and Skin Diseases | AR055299 | Rita CR Perlingeiro |
| National Institute of Arthritis and Musculoskeletal and Skin Diseases | AR071439 | Rita CR Perlingeiro |
| LGMD2I Research Fund | | Anne G Bang<br>Rita CR Perlingeiro |
| Ministerio de Ciencia Tecnología y Telecomunicaciones de Costa Rica | PED-107-2015-2 | Carolina Ortiz-Cordero |

The funders had no role in study design, data collection and interpretation, or the decision to submit the work for publication.

### Author contributions

Carolina Ortiz-Cordero, Conceptualization, Formal analysis, Funding acquisition, Validation, Investigation, Visualization, Methodology, Writing - original draft, Writing - review and editing; Alessandro Magli, Investigation, Writing - review and editing; Neha R Dhoke, Taylor Kuebler, Sridhar Selvaraj, Nelio AJ Oliveira, Haowen Zhou, Investigation; Yuk Y Sham, Anne G Bang, Supervision, Writing - original draft; Rita CR Perlingeiro, Conceptualization, Supervision, Funding acquisition, Writing - original draft, Project administration, Writing - review and editing

### Author ORCIDs

Carolina Ortiz-Cordero (iD) https://orcid.org/0000-0001-6953-0366
Alessandro Magli (iD) http://orcid.org/0000-0003-3874-2838
Rita CR Perlingeiro (iD) https://orcid.org/0000-0001-9412-1118

### Ethics

Animal experimentation: Animal experiments were carried out according to protocols (protocol ID: 2002-37833A) approved by the University of Minnesota Institutional Animal Care and Use Committee.

### Decision letter and Author response

Decision letter https://doi.org/10.7554/eLife.65443.sa1
Author response https://doi.org/10.7554/eLife.65443.sa2

## Additional files

### Supplementary files

• Transparent reporting form

### Data availability

Complete Images for blots and analyzed data is available at Dryad, Dataset, (https://doi.org/10.5061/dryad.x3ffbg7hx).

The following dataset was generated:

| Author(s) | Year | Dataset title | Dataset URL | Database and Identifier |
|---|---|---|---|---|
| Ortiz-Cordero C, Magli A, Dhoke N, Kuebler T, Oliveira NAJ, Zhou H, Sham YY, Bang AG, Perlingeiro RCR | 2021 | NAD+ enhances ribitol and ribose rescue of α-dystroglycan functional glycosylation in human FKRP-mutant myotubes | http://dx.doi.org/10.5061/dryad.x3ffbg7hx | Dryad Digital Repository, 10.5061/dryad.x3ffbg7hx |

The following previously published dataset was used:

| Author(s) | Year | Dataset title | Dataset URL | Database and Identifier |
|---|---|---|---|---|
| Kuwabara N | 2020 | Crystal structure of FKRP in complex with Ba ion, CDP-ribtol, and sugar acceptor | https://www.rcsb.org/structure/6KAM | RCSB Protein Data Bank, 6KAM |

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
