## [Decision Letter]

**Acceptance summary:**

The paper examines several questions regarding a severe form of muscular dystrophy caused by mutations of an enzyme necessary for forming the proper inter-molecular associations that anchor muscle cells to the meshwork of fibers within which they reside. This work elucidates a simple treatment with nutritional supplements using cell culture models. It will be exciting to see how this knowledge translates in other experimental systems.

**Decision letter after peer review:**

Thank you for submitting your article "NAD+ enhances ribitol and ribose rescue of α-dystroglycan functional glycosylation in FKRP-mutant myotubes" for consideration by *eLife*. Your article has been reviewed by two peer reviewers, including Christopher Cardozo as the Reviewing Editor and Reviewer #1, and the evaluation has been overseen by Mone Zaidi as the Senior Editor.

Summary:

In this manuscript, the authors used myotubes derived from iPS cells generated from fibroblasts from a patient with Walker-Warburg syndrome to test the therapeutic potential of pentose phosphate pathway metabolites alone or combined with NAD+ to mitigate the signature biochemical deficit of this disorder, incomplete glycosilation of α-dystroglycan resulting in impaired binding of this protein to laminin. The syndrome is caused by mutations of fukutin-related protein which result in reduced enzymatic activity. The paper biochemically validates this model through its reduced IIH6 99 and laminin binding, and rescue by FKRP transgene introduction. The authors investigate the effect of manipulations of the pentose phosphate shunt through ribose at varying concentrations, its product ribitol and NAD on rescuing the phenotype. The latter is reflected in glycosylation of α-dystroglycan and laminin binding. The rescue was associated with increased ribitol-5-P and CDP ribitol. The rescue, again assayed through reduced IIH6 99 and laminin binding was also enhanced by NAD+, which appeared to act through enhanced ribose, independent of ribitol-5-P and CDP-ribitol. It was compromised by FKRP knockout in FKRP-C318CY myotubes. The paper concludes with possible and reasonable translational suggestions and speculation strongly inviting clinical investigation.

Essential revisions:

The authors are encouraged to consider the following suggestions to improve clarity of the presentation of the data.

Figure 5C: Confirm in the figure or figure legend that experiments done with FKRP KO cells?

Figure 1—figure supplement 1. It seems appropriate to add a description of how karyotyping was done or provide a link to prior literature.

Figure 2—figure supplement 1A and D: it was difficult to discern what is shown in these images (phase contrast, MyHC-immunostaining, or others as appropriate).

---

## [Author Response]

Essential revisions:The authors are encouraged to consider the following suggestions to improve clarity of the presentation of the data.Figure 5C: Confirm in the figure or figure legend that experiments done with FKRP KO cells?

We thank the reviewers for confirming this point. The experiments in Figure 5C were performed using the FKRP KO myotubes. We have revised this figure to make sure this information is clearly conveyed.

Figure 1—figure supplement 1. It seems appropriate to add a description of how karyotyping was done or provide a link to prior literature.

Cytogenetic analysis: Live iPS cells were submitted to the Cytogenomics Core at the University of Minnesota Masonic Cancer Center for G-band karyotype analysis. Cells were treated with colcemid for 3 hours to arrest cells and 20 different metaphases were analyzed at a resolution of 400–450 band level.”

Figure 2—figure supplement 1A and D: it was difficult to discern what is shown in these images (phase contrast, MyHC-immunostaining, or others as appropriate).

We thank the reviewers for bringing this up. We have revised the figure legend to clarify that images represent phase contrast, as shown below.

“Figure 2—figure supplement 1. Ribitol and ribose dose-response studies in iPS cell-derived myotubes.[…] (E) Western blot shows IIH6 immunoreactivity. β-DG was used as loading control. Scale bar, 200μm.”